# Mosquito-independent milk-associated transmission of zoonotic Wesselsbron virus in sheep

Marta Zimoch[1,2,3,4], Llorenç Grau-Roma[2,5], Matthias Liniger[1,2], Noelle Donzé[1,2], Aurélie Godel[1,2], Damián Escribano[6,7], Bettina Salome Trüeb[1,2], Paraskevi Pramateftaki[3,8], Sergi Torres-Puig[8], José Joaqín Cerón[6], Volker Thiel[1,2,3], Jörg Jores[3,8], Artur Summerfield[1,2,3], Nicolas Ruggli[1,2], Charaf Benarafa [1,2,3‡]*, Obdulio García-Nicolás[1,2,3‡]*

1 Institute of Virology and Immunology (IVI), Mittelhäusern, Switzerland, 2 Department of Infectious Diseases and Pathobiology, Vetsuisse Faculty, University of Bern, Bern, Switzerland, 3 Multidisciplinary Center for Infectious Diseases, University of Bern, Bern, Switzerland, 4 Graduate School for Cellular and Biomedical Sciences (GCB), University of Bern, Bern, Switzerland, 5 COMPATH, Institute of Animal Pathology, Department of Infectious Diseases and Pathobiology, Vetsuisse Faculty, University of Bern, Bern, Switzerland, 6 Interdisciplinary Laboratory of Clinical Analysis (Interlab-UMU), Veterinary School, Regional Campus of International Excellence 'Campus Mare Nostrum', University of Murcia, Campus de Espinardo s/n, Espinardo, Spain, 7 Department of Animal Production, Regional Campus of International Excellence 'Campus Mare Nostrum', University of Murcia, Espinardo, Spain, 8 Institute of Veterinary Bacteriology, Department of Infectious Diseases and Pathobiology, Vetsuisse Faculty, University of Bern, Bern, Switzerland

‡ These authors are co-senior authors on this work.
* charaf.benarafa@unibe.ch (CB); obdulio.garcia-nicolas@ivi.admin.ch (OG-N)

**Data Availability Statement:** The authors confirm that all data underlying the findings are fully available without restriction. All relevant data are

## Abstract

Wesselsbron virus (WSLV) is a zoonotic, mosquito-borne orthoflavivirus endemic to sub-Saharan Africa, causing abortions and stillbirths in small ruminants. The life cycle of WSLV involves Aedes mosquitoes and various wildlife and domestic animals. Seminal studies in the 1950s have shown the zoonotic potential of WSLV, notably in accidental infections of laboratory workers exposed to infected material. More recent epidemiological studies suggest the emergence of clade I WSLV strains in peri-domestic and rural areas of western and eastern Africa. The pathobiology of recent clade I WSLV strains is unknown and no virus isolate is available. To address these gaps, we generated a recombinant clade I WSLV SA999 infectious clone (rSA999) by reverse genetics. Subsequently, lactating ewes were inoculated intravenously with the WSLV rSA999 strain or the clade II SAH177 strain in insect-free biocontainment stables. Inoculated ewes developed fever, viremia, and showed high levels of viral RNA at mucosal surfaces, and elevated viral titers in milk. Milk production was reduced, which directly affected the growth of the lambs, particularly within the rSA999 group. The ewes with higher WSLV titers in their milk in each group transmitted the infection to their lambs, which developed fever, prolonged viremia, and virus secretion. All infected animals produced high antibody titers with cross-neutralizing activity against both WSLV strains. Histopathology and blood biochemistry analysis indicated liver damage associated with necrotizing hepatitis lesions and active viral replication in some cases, which was more pronounced in the rSA999 group. Notably, only the SAH177-infected animals exhibited

within the paper and its Supporting Information files.

**Funding:** This work was principally funded by a grant from the Multidisciplinary Center for Infectious Diseases (MCID), University of Bern (Grant MA_17 to CB and OG-N; https://www.mcid. unibe.ch/research/mcid_funded_projects/index_ eng.html). This work was supported in part by the Swiss National Science Foundation (grant 192498 to AS; https://www.snf.ch). The synthetic genomics work was supported by the Multidisciplinary Center for Infectious Diseases (MCID), University of Bern (Grant MCID_BPBB to JJ). The funders had no role in study design, data collection and analysis, decision to publish, or preparation of the manuscript.

**Competing interests:** The authors have declared that no competing interests exist.

lesions consistent with meningoencephalitis, suggesting that WSLV clade II strains are neurotropic and that clade I strain are more hepatotropic. These findings demonstrate a previously unrecognized mode of vector-free transmission of WSLV that raises significant concerns for public and animal health.

## Author summary

Wesselsbron virus (WSLV) was first described in the 1950s as a cause of abortions in sheep and goat herds in South Africa. The zoonotic potential of this neglected orthoflavivirus became evident in early studies, where laboratory workers developed disease following direct contact with infected samples. Currently, two genetic clades of WSLV are circulating in sub-Saharan Africa: clade I strains are widely distributed and reported more frequently than clade II strains, which are confined to South Africa. The pathogenicity and mode of transmission of more recent WSLV strains have not been studied. In this study, we established a reverse genetics method to generate a recombinant clade I WSLV SA999 infectious clone (rSA999). To explore direct transmission, lactating ewes were inoculated with either the WSLV rSA999 strain or the clade II SAH177 strain. All ewes developed fever, viremia, and showed high levels of viral RNA at mucosal surfaces and milk. Several lambs (40%) in each group developed severe disease, and WSLV transmission to the lambs correlated with the highest viral titers in the milk of their respective mothers. All infected animals produced cross-neutralizing antibodies against both WSLV strains. Histopathological and blood biochemistry analyses suggested a strain-specific predominant tropism for liver (clade I) and brain (clade II). Overall, these findings highlight a new mode of vector-free transmission of WSLV that may pose a risk for newborn humans and animals consuming raw milk products.

## Introduction

Since the beginning of the 21st century, more than 10 major epidemics of viral diseases have been documented, with many originating from animals [1]. The increased interactions between humans and animals that contribute to such epidemics are driven by human activities such as incursions into natural habitats and their destruction, wildlife poaching, international travel, and intensive farming. With respect to vector-borne pathogens, adaptation of insect vectors to urban/peridomestic environments and climate change contribute to the spread of arthropod vectors into new geographical areas, thereby increasing the possibility for diseases to spread into naïve host populations [2–4]. The family of *Flaviviridae* includes vector-borne viruses with significant pandemic potential, with virus reservoirs found in both wild and domestic animals [1,5,6]. For instance, the outbreaks of Zika virus in South America and West Nile virus in North America and Europe were not anticipated [7–9]. Furthermore, certain flaviviruses, such as tick-borne encephalitis virus and Japanese encephalitis virus, can be transmitted even in the absence of an arthropod vector [10,11]. Therefore, transmission studies of agents of emerging zoonotic diseases foster preparedness against such threats.

Wesselsbron (WSL) disease is a neglected mosquito-borne illness caused by Wesselsbron virus (WSLV), an orthoflavivirus phylogenetically related to yellow fever virus. WSLV was firstly isolated from the liver of an aborted lamb following an outbreak in a farm in South Africa [12]. WSLV is transmitted by *Aedes* spp. mosquitoes and causes disease principally in

sheep and goats [13,14]. Natural outbreaks of WSL disease are clinically similar to Rift Valley fever with abortions and death of newborn lambs, which requires appropriate sampling and multiple techniques to confirm the diagnosis. For neglected orthoflaviviruses like WSLV, the role of different vertebrate species in the virus' life cycle remains poorly understood [15]. WSLV specific antibodies or the virus itself were reported from many domestic and wildlife animals including cattle, pigs, horses, camels, dogs, farmed ostriches, zebras, short-eared gerbils, and commensal black rats [12,16–18]. The seroprevalence in humans was reported to be up to 30% in specific African regions [19–21]. However, interpretation of serum reactivity data should be cautious due to potential antibody cross-reactivity between orthoflaviviruses [18]. Several cases of WSL disease were reported in humans presenting with fever, myalgia, arthralgia, and headache, but in the absence of proper diagnostics natural infections are likely to be under- or mis-diagnosed [12,21–23]. Infections of laboratory workers following occupational exposure were reported following the handling of virus isolates in veterinary laboratories, necropsies, and mosquito preparations in the early studies describing WSLV. The circumstances of infection strongly suggest direct transmission as shown in a case of encephalitis following splashing of contaminated material in the eye [21].

Experimental WSLV infections in pregnant ewes result in still birth and abortions with fetus malformations [12,24]. In adult domestic sheep, WSLV typically induces an acute fever, sometimes biphasic, with a wide range of symptoms from a mild fatigue to death [12,25,26]. In newborns, the disease is acute with fever, anorexia, lethargy, increased respiratory rate, and a mortality rate up to 30% [14]. These studies were principally carried out in African sheep breeds using clinical isolates passaged in newborn mouse brains and genome sequence data is limited or absent. Genetic analysis of WSLV strains based on NS5 sequence reveals that WSLV can be classified in two clades: clade I that is found in many regions of sub-Saharan Africa, and clade II that are only found in South Africa in areas overlapping with clade I [21]. A more recent study using the clade II strain SAH177, which was originally isolated from a human and passaged in mouse brains and Vero cells [27], reported early pathogenesis of WSLV infection in pregnant ewes [28]. In the latter study, experimentally infected ewes showed no fever and only a transient viremia for the first 4 days post infection (dpi). Surprisingly, abortion was absent up to 8 dpi, when the necropsies were performed. Viral RNA was only found in fetal organs, cord blood and the placenta, but not in the other tissues from the ewes [28]. The pathobiology of clade I WSLV strains has not been studied since the seminal description of WSL disease by Weiss *et al.* (van Tonder strain) [12]. Experimental inoculation was the most severe and fatal in newborn lambs, while the disease was milder and resolved in older lambs. A large cohort of pregnant ewes were then infected intravenously with the van Tonder strain leading to birth of fatally sick animals, abortions and death of ewes all within a week of infection, when the experiment was terminated because of biosafety concerns [12].

To address knowledge gaps in the transmission and pathobiology of WSLV and to expand experimental research on contemporary strains of WSLV, we developed a reverse genetic approach to generate infectious WSLV clones and investigated the pathobiology of clade I and II strains of WSLV in a European sheep breed. We found that lactating ewes infected with recombinant clade I WSLV strain SA999 (rSA999) had more severe liver damage compared to those infected with SAH177, but all the ewes recovered. WSL disease induced by both strains led to reduced growth rates in lambs, and the virus was transmitted to suckling lambs in the absence of insect vectors, leading to severe disease and demonstrating a new mode of WSLV transmission.

## Results

### Generation of clade I WSLV rSA999 strain by reverse genetics

We employed transformation-associated recombination (TAR) cloning in yeast to generate a recombinant clone of the clade I WSLV SA999 strain based on published full-length sequence (GenBank MK163943). Six overlapping subgenomic fragments encompassing the entire viral genome were chemically synthetized and transformed together with the cloning vector pCC1BAC-his3 into *Saccharomyces cerevisiae* (S1 Fig). A correctly assembled and sequence-confirmed clone was used for *in vitro* transcription and 5' capping; the mRNA was transfected in Vero cells and supernatants were passaged twice on C6/36 cells to generate rSA999 virus working stock at passage 2. No variant was identified in the sequence of the rSA999 working stock compared to the reference sequence (S1 Appendix). In contrast, three variants were identified in the SAH177 working stock relative to the reference sequence (GenBank: DQ859058) [29] (S2 Appendix).

### Pathogenicity of clade I and II WSLV in lactating ewes and their lambs

To evaluate the pathogenicity and transmission of clade I and clade II WSLV in sheep, lactating ewes with their respective lambs were successively allocated to the mock group (n = 3), then to one and the other group to be inoculated. The groups were housed in three different BSL3Ag biocontainment stables. Five ewes were inoculated intravenously with $10^5$ TCID$_{50}$ of recombinant WSLV clade I (rSA999), another 5 ewes were inoculated with the same infectious dose of WSLV clade II (SAH177), and 3 ewes were mock-inoculated with C6/36 cell culture supernatant (Fig 1A). The lambs were not subjected to experimental inoculation. Following infection with rSA999, all ewes presented a simultaneous onset of fever and clinical manifestations at 2 days post-infection (dpi), with varying recovery periods and fever duration ranging from 2 to 6 days. Within the lambs of this group, two of the 5 developed high fever and clinical symptoms around 8 dpi, with one of them becoming critically ill and died on 11 dpi (Fig 1B and 1C). In comparison, the onset of disease in SAH177-infected ewes was slightly delayed, occurring between 2–4 dpi. Clinical signs or fever persisted for 3 to 9 days, except for one ewe whose body temperature never reached 40.5˚C. Within this group, 3 out of 6 lambs showed fever and clinical signs of disease at 5–6 dpi, which resolved by 11 dpi (Fig 1D and 1E). Clinical manifestations in both infected groups were nonspecific and included fever, reduced liveliness, loss of appetite, and dyspnea; no specific neurological signs were observed. Lambs with fever developed the same clinical signs as the ewes. Animals in the mock group showed no clinical signs and normal body temperature throughout the study (Fig 1F). Daily measurements of the lamb's body weight revealed significantly lower weight gain in both infected groups compared to the mock group, with lambs from the rSA999-infected ewes being significantly more affected than those from the SAH177 group (Fig 1G).

### Kinetics of WSLV viremia in ewes and lambs

Viremia was detected in the majority of the inoculated ewes on 2 dpi, except for one ewe in the rSA999 group with viral RNA detection in serum on 1 dpi, and another on 3 dpi in the SAH177 group. In the rSA999 group, viremia peaked at 3 dpi with high virus load ($>10^7$ copies/ml) in most ewes, resolving within 6 dpi for 3 ewes, while two animals still had detectable viremia at the end of the study at 12 dpi. Viremia was observed from 7 dpi in both rSA999 lambs that showed clinical disease with high peaks ($>10^8$ copies/ml) at 8–9 dpi, and viremia remained detectable until the final sampling (Fig 2A). In the SAH177 group, viremia peaked in ewes at 3–4 dpi and resolved within 4–6 dpi. The 3 lambs with clinical disease also had

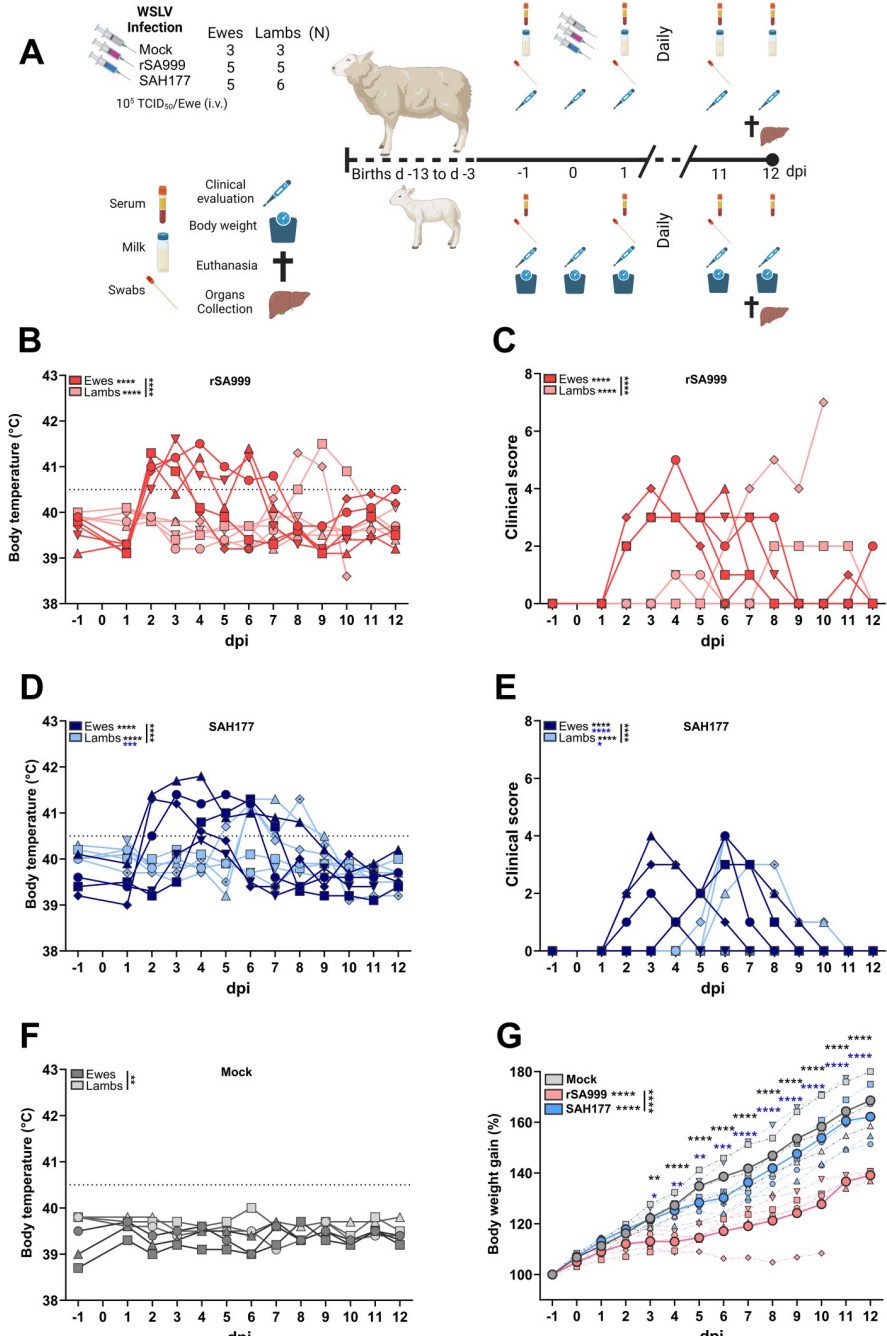

**Fig 1. Wesselsbron disease kinetics in inoculated ewes and their lambs.** (**A**) Experimental design of the animal study. Lactating ewes and their lambs were randomly allocated to 3 groups and intravenously inoculated with 1ml of cell culture supernatant (mock), or $10^5$ $TCID_{50}$ of WSLV strains from clade I (rSA999) or clade II (SAH177). Daily monitoring and sampling included body temperature, clinical score, swabs, and blood samples from all animals, and milk collection from ewes and body weight of the lambs. (**B,C**) Body temperature (**B**) and clinical score (**C**) of ewes and lambs of the rSA999 group. (**D,E**) Body temperature (**D**) and clinical score (**E**) of ewes and lambs of the SAH177 group. (**F**) Body temperature of ewes and lambs of the mock group. (**G**) Relative body weight gain of the lambs of the 3 groups in percent of initial weight at day -1 prior to inoculation. Distinct symbols represent different animals throughout the manuscript within each group; lambs have the same symbol as their mother; twin lambs (SAH177 group) are distinguished by a dotted symbol. Differences between groups were analyzed by one-way ANOVA with Tukey post hoc test for comparing the area under the curve (AUC), and Mann-Whitney U test for daily individual differences in body weight. Underlined asterisks indicate statistical significance between groups showed in the same panel; asterisks next a group label indicates statistical significance compared to the mock group of the same age; blue

asterisks indicate statistical differences between the rSA999 and the SAH177 groups; in (**G**) asterisks positioned on each day show statistical differences between rSA999 and mock (black asterisks) or with the SAH177 (blue asterisks) groups. $p<0.05$ was considered statistically significant (*$p<0.05$; **$p<0.01$; ***$p<0.001$; **** $p<0.0001$). Panel A was created with BioRender.

detectable viremia on 5 dpi and up to 8–10 dpi (Fig 2B). On average, the peak of viremia was higher in ewes inoculated with rSA999 and the area under the curve (AUC) was significantly greater in the rSA999 group. The 5 lambs with detectable viremia were from 2 ewes in each group that reached peak viremia $>10^7$ copies/ml. Of note, one ewe in the SAH177 group (rhomboid symbol) had twin lambs, both of which became viremic.

## High WSLV titers in milk of infected ewes correlates with transmission to lambs

WSLV was detected in the milk of all ewes except for one SAH177-infected ewe, which exhibited a particularly brief viremia period (Fig 2C and 2D, square symbol). Interestingly, viral RNA remained detectable in milk when it was no longer detectable in serum, suggesting local replication or specific retention of viral particles in the mammary gland. In the 4 ewes that transmitted the virus to their lambs, high viral RNA ($>10^4$ copies/ml) was measured for 3 consecutive days from 4 dpi in rSA999-infected ewes and for 6 consecutive days from 3 dpi in SAH177 ewes (Fig 2C and 2D). Milk samples collected at 5 dpi were titrated on C6/36 cells, confirming the presence of infectious virus in the 9 ewes that tested positive by RT-qPCR (Fig 2E). Indeed, there was a strong correlation between the viral load determined by RT-qPCR and the infectious virus determined by $TCID_{50}$ in the same samples on the same day (Fig 2F). Moreover, the lambs that became infected with WSLV were from the 4 ewes with the highest titers ($>10^4$ $TCID_{50}$/ml) in their milk suggesting a potential route of infection as ewes typically only allow their own lamb to suckle. Importantly, the lamb from the only ewe without detection of WSLV in the milk remained negative. WSLV was also detected by qPCR in nasal and rectal swabs of most of the infected ewes, exhibiting patterns similar to those of viremia (Fig 2G and 2H). In contrast, WSLV was only sporadically detected in ocular and oral swabs of a few ewes in both groups (Fig 2I and 2J). Variability in daily copies in some nasal and ocular conjunctival samples may in part be explained by the alternate sampling of the left and right sides. Overall, ewes infected with rSA999 had higher viral RNA copies in nasal, oral, ocular swabs than those inoculated with SAH177. No significant differences between groups were observed in rectal swab virus load. WSLV was detectable by RT-qPCR in oronasal swabs of all the lambs that developed clinical signs of disease (Fig 2K).

## Fast generation of neutralizing antibodies

Neutralizing antibodies were rapidly detected in all inoculated ewes. Remarkably high serum dilutions ($ND_{50}> 1:36'200$) revealed neutralizing activity from 5 dpi in the rSA999 ewes (Fig 3A) and from 5–6 dpi in SAH177 ewes (Fig 3B). Seroconversion and high neutralizing antibody levels were only observed in the lambs with detectable viremia, indicating active seroconversion in these lambs rather than passive transmission of neutralizing antibodies through milk. As expected, WSLV and serum neutralizing activity were undetectable in the ewes and lambs of the mock group. Neutralizing antibody titers were also rapidly detectable in milk from 5 dpi in the 2 ewes (rSA999 group) that had the earliest onset of serum antibodies, with all ewes displaying very high neutralizing antibody activity in milk by 10 dpi (Fig 3C). Furthermore, antibodies induced by the rSA999 or SAH177 strains had a significant cross-reactivity with the heterologous WSLV strain (Fig 3D).

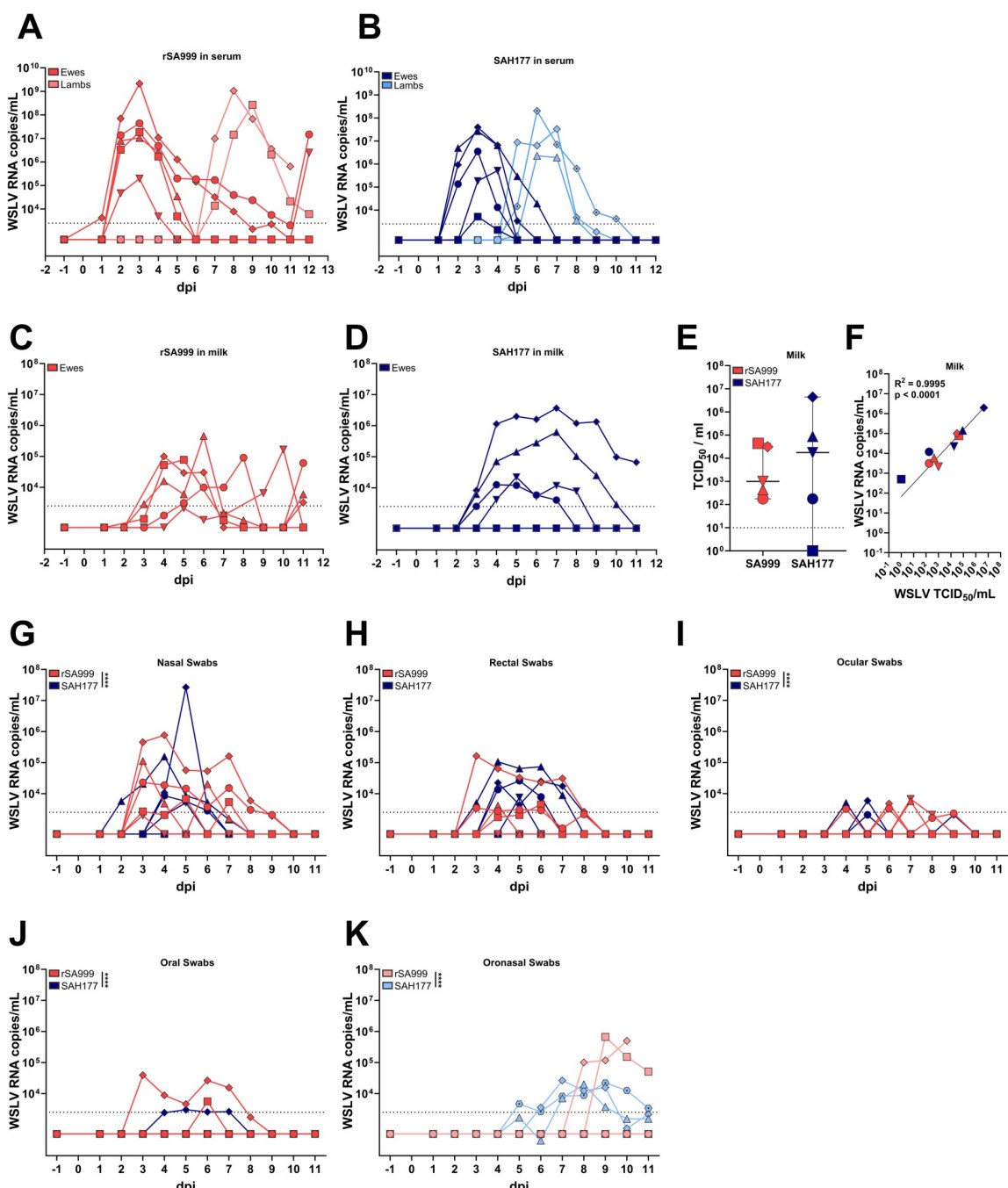

**Fig 2. WSLV RNA load in blood, milk, and mucosal surfaces.** (**A**,**B**) Viremia determined by RT-qPCR in serum samples ewes and lambs of the rSA999 (**A**) and SAH177 (**B**) groups. (**C**,**D**) Viral RNA determined by RT-qPCR in milk of infected ewes of the rSA999 (**C**) and SAH177 (**D**) groups. (**E**) Infectious WSLV titers in milk samples at 5 dpi determined by end-point dilution and expressed as $TCID_{50}$/ml. (**F**) Correlation analysis between RNA copies and infectious viral particles in the milk of infected ewes. (**G-J**) WSLV RNA copies in mucosal swabs from ewes: nasal (**G**), rectal (**H**), ocular (**I**), and oral (**J**). (**K**) WSLV RNA copies in oronasal swabs from lambs. Distinct symbols are used to represent different animals; within each group, lambs are denoted by the same symbol as their respective mother. Differences between groups were analyzed by one-way ANOVA with Tukey post hoc test for comparing the area under the curve (AUC). Underlined asterisks indicate statistical significance between groups showed in the same panel. $p < 0.05$ was considered statistically significant (*$p < 0.05$; **$p < 0.01$; ***$p < 0.001$; ****$p < 0.0001$).

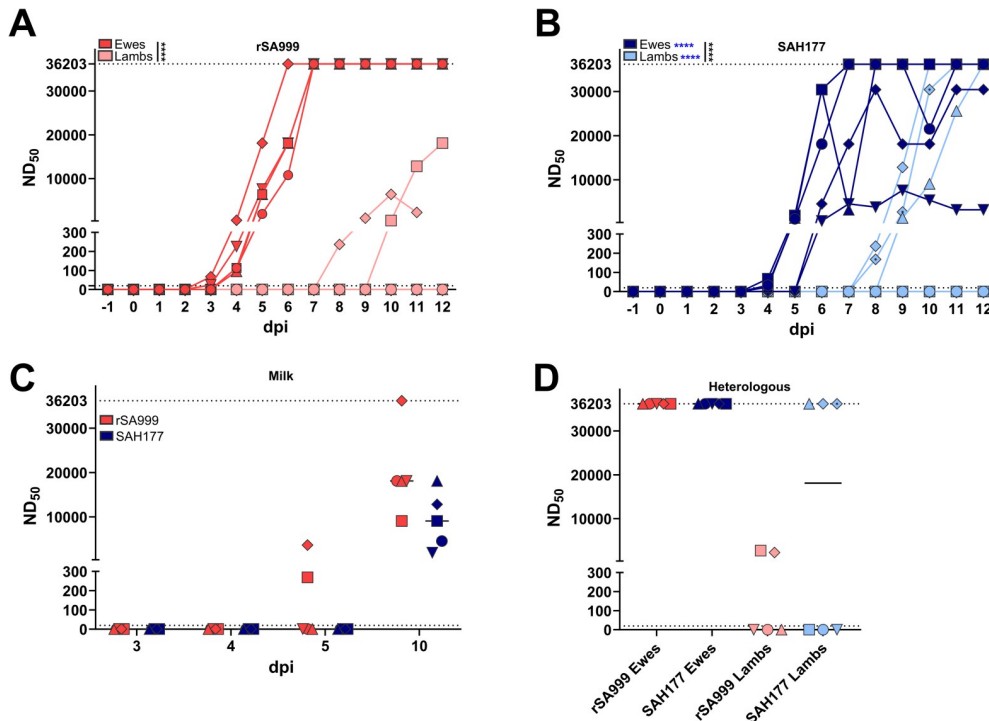

**Fig 3. Neutralizing antibodies against WSLV during the course of infection.** (**A,B**) Neutralizing antibody titers in daily serum samples of the rSA999 (**A**) and SAH177 (**B**) groups. The mock group had no detectable neutralizing antibodies throughout the study. (**C**) Neutralizing antibody titers in milk at indicated time points. (**D**) Neutralizing serum antibody titers of indicated groups at 10 dpi against the heterologous virus. The neutralizing dose 50% ($ND_{50}$) was determined using two-fold serial dilution of serum starting at 1:10 mixed with 200 FFU of WSLV. Distinct symbols are used to represent different animals; within each group, lambs are denoted by the same symbol as their respective mother. (**A, B**) Differences between groups were analyzed by one-way ANOVA with Tukey post hoc test for comparing the area under the curve (AUC). Underlined asterisks indicate statistical significance between groups showed in the same panel; blue asterisks indicate statistical differences in viremia between the rSA999 and the SAH177 groups. $p < 0.05$ was considered statistically significant (****$p < 0.0001$).

## Viral load in organs

At the end of the study at 12 dpi, all the ewes infected with WSLV rSA999 had detectable virus by RT-qPCR in spleen, tonsils, and liver (Fig 4H, 4E and 4I), which contained the highest viral load. In addition, WSLV was detected in lymph nodes and kidneys of 3–4 ewes in this group (Fig 4F, 4G and 4J). In the central nervous system (CNS, Fig 4A–4D), few samples had detectable virus by RT-qPCR, including olfactory bulb, cortex, cerebellum, and thalamus. In the mammary gland, WSLV was only detected in one ewe infected with rSA999 (Fig 4K). In ewes inoculated with SAH177, the liver was the only organ where the virus persisted in all animals at the end of the study (Fig 4I). In the other organs, the detection of viral RNA was highly variable: one ewe had detectable virus in most tissues (blue rhombus), while the ewe with the shortest viremia (blue square) was RT-qPCR negative in all other organs. Few organs were positive for virus RNA in the three other ewes (Fig 4A–4K).

In the viremic lambs of both groups, WSLV was detected in all tested organs, with the highest virus loads found in the liver ($10^6$–$10^{10}$ copies/g; Fig 4I), while the thymus and the CNS presented the lowest viral loads (; Fig 4A–4D and 4L). The lamb from the rSA999 group that perished at 11 dpi had a different pattern, with high viral loads ($>10^6$ copies/g) found in the liver, CNS, spleen, kidney, and lymph nodes (Fig 4, pink rhombus). The viral RNA was not detectable in any organ of the lambs that remained healthy. As expected, no virus was

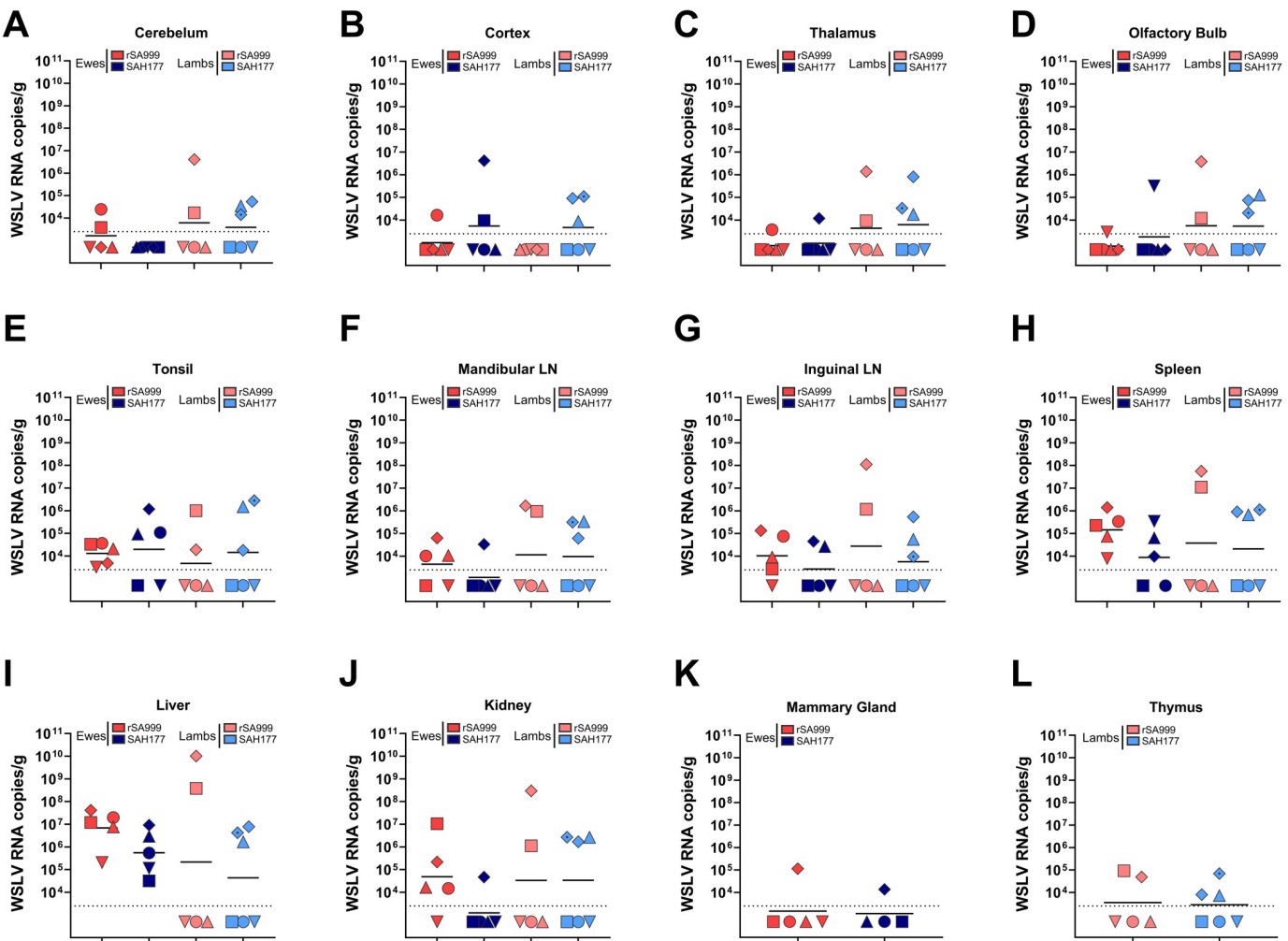

**Fig 4. WSLV viral RNA loads in organs.** WSLV RNA levels were determined using RT-qPCR in the indicated organs of ewes and lambs collected at necropsy at 12 dpi, except for 1 ewe (red rhomboid) and its lamb (light red rhomboid), which were collected at 11 dpi as the lamb succumbed from the infection. The absolute WSLV RNA levels are expressed as copies per gram of tissue. Distinct symbols are used to represent different animals; within each group, lambs are denoted by the same symbol as their respective mother. No statistical analysis was performed.

detectable in the tissue samples from ewes and lambs in the mock group. Overall, the liver appears as the most reliable organ to sample for WSLV diagnostic purposes regardless of the age of the animals and the clade.

## Clinical biochemistry

During the experiment, the serum from healthy animals presented a light-yellow tone. However, as clinical signs developed, the serum color intensified, showing hues of dark yellow/orange in the most severely affected animals. Jaundice is a result of elevated bilirubin levels in the blood, typically associated with liver disease. To further investigate the pathobiology of WSLV infection, clinical biochemistry markers were measured in the daily plasma samples. Increased levels of aspartate aminotransferase (AST), bilirubin, bile acids, and adenosine deaminase (ADA) levels were measured one day after viremia was detected in inoculated ewes (Fig 5A–5D), which were more pronounced in ewes and lambs infected with the rSA999 strain compared to the SAH177 strain. Furthermore, a steady increase in cholesterol levels was

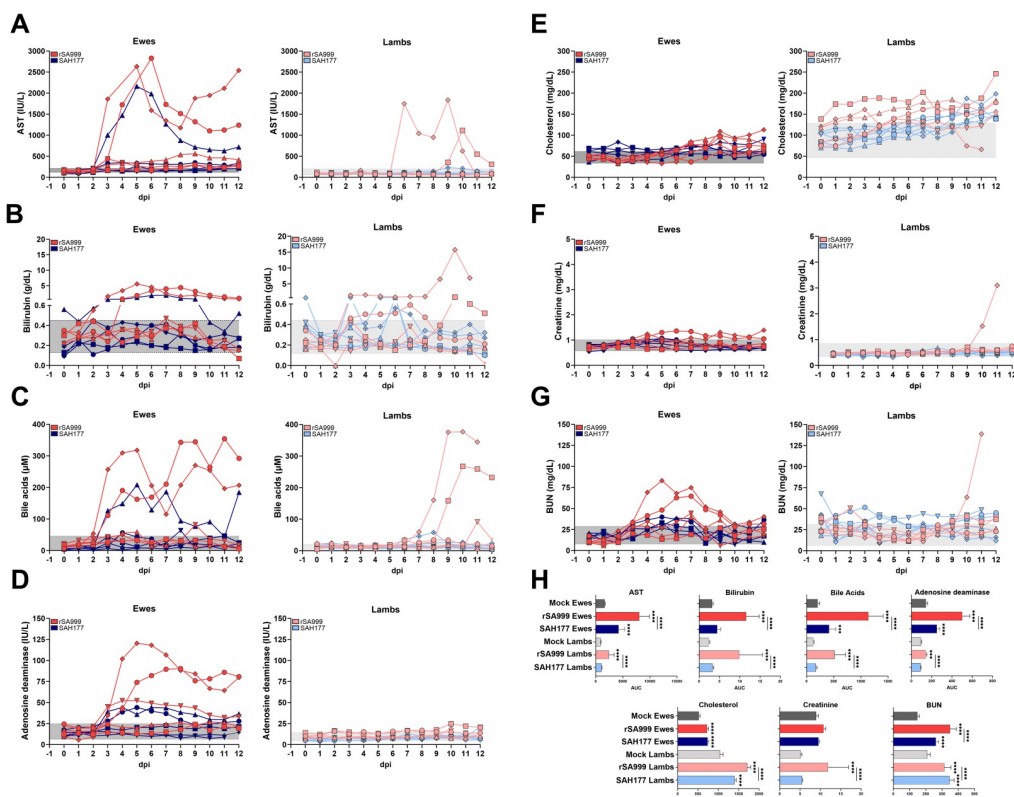

**Fig 5. Serum clinical biochemistry parameters in WSLV-infected sheep.** Serum samples collected daily were inactivated with 0.5% v/v NP-40 were analyzed for biomarker concentrations, including (**A**) aspartate aminotransferase (AST), (**B**) bilirubin, (**C**) bile acids, (**D**) adenosine deaminase, (**E**) cholesterol, (**F**) creatinine, and (**G**) blood urea nitrogen (BUN). The maximum and minimum values from the mock group of ewes and lambs were indicated for reference in gray shading background. (**H**) Group differences were analyzed by one-way ANOVA of the AUC with Tukey *post hoc* test. An asterisk on a specific group indicates differences compared to the age-matched mock; underlined asterisks indicate differences between infected groups. Statistical significance was considered for $p < 0.05$ (***$p < 0.001$; **** $p < 0.0001$).

observed overtime, also reflecting decreased liver function (Fig 5E). Concurrent increase in creatinine and urea nitrogen (BUN) levels indicative of renal failure was observed in the animals with the highest viremia (Fig 5G) and were especially high in the most severely affected lamb (Fig 5G). Other biomarkers were also altered further indicating liver disease and metabolic disturbance such as alkaline phosphatase (ALP), albumin, triglycerides, lactate, and total proteins (S2 Fig).

## Pathology and immunohistochemistry

Euthanasia and postmortem examination performed at 12 dpi revealed only few mild macroscopic lesions. The recorded changes included clear serous edema (20–100 ml) in several body cavities, including mild ascites and hydrothorax in two ewes (one from each group), and mild hydropericardium (two ewes from the SAH177 group). In addition, several ewes from all groups, including the mock, showed mild to moderate liver fluke infestation and mild verminous pneumonia.

Histologically, 5 animals (3 ewes and 2 lambs) from rSA999 group and 6 animals (3 ewes and 3 lambs) from SAH177 group showed multifocal, random necrotizing hepatitis (Fig 6A and 6B). These hepatic lesions were mild in all animals from the SAH177 group, while the two

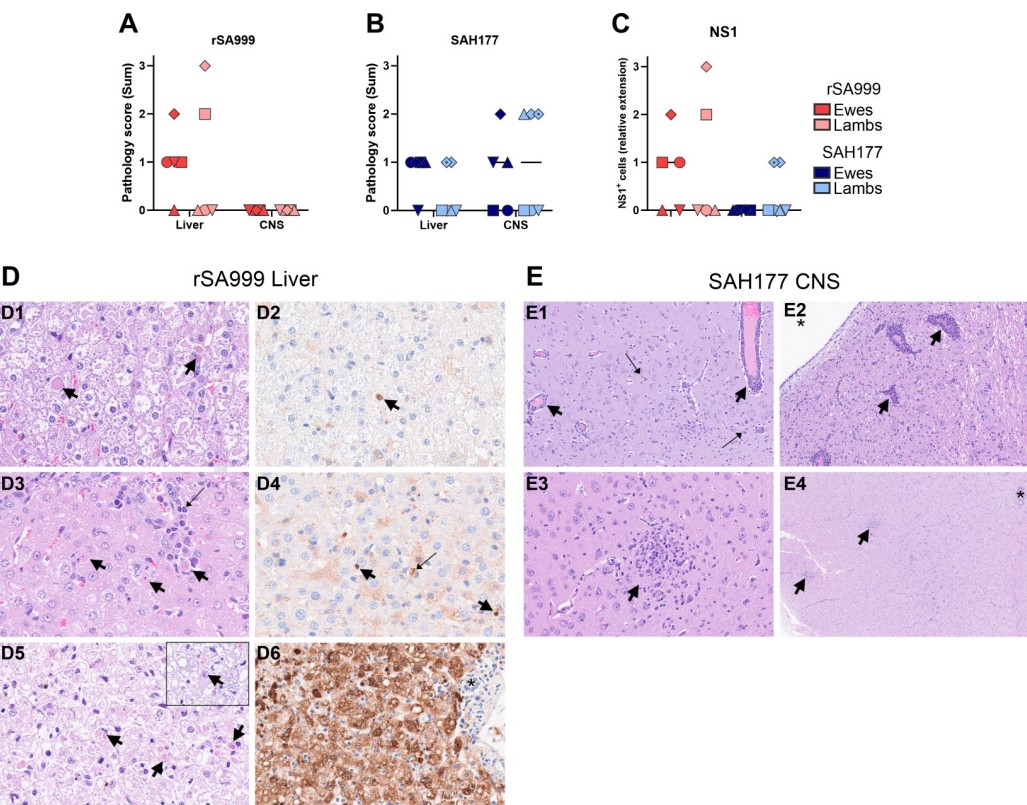

**Fig 6. Histopathology and immunohistochemistry of WSLV NS1 in liver and brain samples.** (**A**,**B**) Semiquantitative scores of necrotizing hepatitis (liver) and non-suppurative meningoencephalitis (CNS) of ewes and lambs from rSA999 (**A**) and SAH177 (**B**) groups as none (0), 1 (mild), 2 (moderate) and 3 (severe). (**C**) Semiquantitative score of immunohistochemistry (IHC) positivity for WSLV NS1 in liver samples as negative (0), positive <5% of hepatocytes (1), 5–20% of hepatocytes (2) and >20% of hepatocytes (3). (**D**) Microscopic findings in the liver of animals infected with WSLV rSA999. (**D1,3,5**) H&E stains showing hepatic lesions of mild (**D1**), moderate (**D3**) and severe (**D5**) intensity. (**D2,4,6**) IHC for WSLV NS1 with low (**D2**), moderate (**D4**), and marked (**D6**) positivity. (**D1**) Ewe #2 (red square in panels 6A,C), multifocal hypereosinophilic, shrunken hepatocytes resembling Councilman bodies (arrows). (**D2**) Multifocal coarsely granular NS1 positivity is present within the cytoplasm of few hepatocytes (arrow). The hepatocytes show diffusely a mild intracytoplasmic vacuolar degeneration. (**D3**) Lamb #L2 (pink square in panels 6A,C), multifocal hypereosinophilic, shrunken hepatocytes with pyknosis and karyorrhexis (bold arrows), lymphocytic infiltrates (thin arrow) are scattered within the parenchyma. (**D4**) Multifocal coarsely granular NS1 positivity is present within the cytoplasm of hepatocytes (bold arrows) and Kupffer cells (thin arrow). (**D5**) Lamb #L5 (pink rhombus), there is diffuse loss of hepatic architecture, the hepatocytes show prominent intracytoplasmic vacuolar degeneration and multifocally are hypereosinophilic and fragmented (bold arrows). Inset: Round eosinophilic structure with marginated chromatin in the nucleus of a hepatocyte (intranuclear inclusion body) (bold arrow). (**D6**) Strong diffuse intracytoplasmic NS1 positivity in hepatocytes and Kupffer cells; the portal tract on the right of the image including its bile duct (asterisk) is negative. (**E**). Microscopic findings in H&E-stained CNS sections of animals infected with WSLV SAH177. (**E1**) Lamb #L8 (triangle in figure), cerebral cortex with accumulation of inflammatory leukocytes, predominantly lymphocytes, within the perivascular Virchow-Robin space (perivascular cuffs) (arrows) with multifocal neuronal degeneration and necrosis (thin arrows). (**E2**) Lamb #L8 (triangle in figure), brainstem with multifocal lymphocytic perivascular cuffs (bold arrows) adjacent to the lumen of the fourth ventricle (asterisk). (**E3**) Lamb #L10.1 (rhombus in figure), cerebral cortex with focal and prominent proliferation of glial cells (glial nodule, bold arrow) within the cortical grey matter. (**E4**) multifocal glial nodules within the white matter of the cervical spinal cord (bold arrows); the central canal of the spinal cord is on the top right of the image (asterisk).

lambs with hepatic lesions from the rSA999 group showed moderate and severe lesions, respectively. The hepatic lesions were typically characterized by individual or small clusters of hypereosinophilic rounded-off hepatocytes often resembling Councilman bodies, generally accompanied by a predominantly lymphocytic infiltrate with fewer neutrophils and prominent Kupffer cells (Fig 6D). The lamb from the rSA999 group found dead at 11 dpi, despite showing

a moderate status of autolysis, presented severe hepatic lesions characterized by widespread areas of hepatocellular degeneration and necrosis with presence of intranuclear structures compatible with inclusion bodies (Fig 6D, and inset). No lesions of necrotizing hepatitis were observed in the controls. In addition, vacuolar hepatocellular degeneration of varying intensity was observed in most of the animals, which was moderate in four ewes from the rSA999 group and one from the mock group. Furthermore, hyperplastic lymphocytic cholangitis typical of *Dicrocoelium dendriticum* infestation was observed in most ewes. Lesions of non-suppurative meningoencephalitis were only observed in animals from the SAH177 group (3 ewes and 3 lambs). These three lambs were offspring from two ewes with lesions in the CNS (Fig 6B). The meningoencephalitis was of moderate intensity in four animals (3 lambs and 1 ewe) and mild in the remaining 2 ewes. Moreover, two of the animals (1 ewe and 1 lamb) showed concomitant mild to moderate lesions consistent with non-suppurative myelitis in the cervical spinal cord. The CNS lesions consisted of multifocal lymphocytic perivascular cuffs, neuronal necrosis, gliosis, and glial nodules in the grey and the white matter (Fig 6E). No lesions of encephalitis were observed in the animals infected with rSA999 or in the controls. In the kidney, two ewes from group rSA999 showed mild multifocal tubular degeneration and regeneration, while the lamb that succumbed showed extensive tubular degeneration and necrosis, as well as multifocal areas of mineralization. No such changes were observed in the kidney of animals from the SAH177 group. No significant histological changes were observed in the other examined organs. Altogether, the histological lesion scores in liver, although not prominent, were higher in the rSA999 group than in the SAH177 group, while only the latter showed lesions in the CNS, which correlated well with virus loads at this terminal time point (Fig 6A and 6B).

Immunohistochemistry (IHC) for WSLV showed positive staining in the liver of 7 animals, corresponding to 3 ewes and 2 lambs from the rSA999 group and 2 lambs from the SAH177 group (Fig 6C and 6D). The staining was intracytoplasmic within hepatocytes and Kupffer cells, coarsely granular to diffuse, and of moderate to strong intensity. All control samples were negative. No convincing positivity was observed in tissues other than liver. Overall, these findings indicate that WSLV strains from both clades I and II actively replicate in adult sheep and can be equally transmitted to suckling lambs. Moreover, results suggest that the two strains may differ in their tissue tropism preferences: neurologic (SAH177) and hepatic (rSA999).

## Discussion

Research on WSLV was limited until recently by the scarcity of low-passage virus isolates and the lack of full-length WSLV genome sequences. With recent developments in sequencing technologies, it is likely that new genomic data will become available as reported recently [30,31]. We have established an efficient method to generate infectious clones of WSLV using a TAR cloning-based reverse genetics approach, which will allow the study of a broader range of relevant contemporary strains recently sequenced [17,30,31]. The absence of variant in the sequence of virus stocks generated by this recombinant approach is also an advantage of this methodology. Indeed, mutants can be rapidly generated in one or several synthetic DNA fragments and reassembled in yeast to investigate the effects of specific mutations or gene deletions associated with virulence and cellular tropism. Importantly, combined modifications of envelope proteins, disabling mutations in non-structural genes, and one-to-stop mutagenesis strategies, as recently used for a new generation SARS-CoV-2 vaccine [32], provide a platform to develop a safe attenuated vaccine for WSLV and, by extension, for other orthoflaviviruses.

Our in vivo transmission studies have clearly shown that both clade I and II WSLV can be transmitted horizontally by direct contact between animals without insect vectors. WSLV

induced mild to moderate illness with fever in all ewes, associated with a reduction of milk production reflected by the reduced growth rate of the lambs, which was already apparent before any of the lambs were infected. The rate of transmission from ewes to lambs was comparable for the two WSLV strains, with 3 out of 6 lambs in the SAH177 group and 2 out of 5 lambs in the rSA999 group. The lambs that became viremic developed moderate to severe illness, including one death following sudden deterioration. These data are consistent with field observations of WSLV outbreaks in sheep and goats, where abortions in the last third of pregnancy and death of newborn animals were reported [12,14]. However, our data provide the first experimental evidence that lambs can be infected by contact after birth, leading to severe illness in the absence of mosquito vectors. We found very high RNA viral loads in milk for several days, correlating with high infectious titers of WSLV, and 4 ewes had high viral RNA load in milk at the end of the study. This finding is highly relevant for public health regarding the consumption of raw milk and derived products, and future work should determine how long infectious virus is secreted in the milk. Similarly, tick-borne encephalitis virus is secreted in the milk of small ruminants, leading to infections in humans consuming raw milk products [33–35].

Transmission of WSLV to lambs was strongly associated with higher viremia and higher WSLV titers in the milk of the respective ewes in both groups. Our findings are also reminiscent of mother-to-child transmission following vaccination with live attenuated yellow fever and Zika vaccines to breast-feeding infants [36]. However, we cannot formally exclude the possibility that WSLV can also be transmitted via exposure to other body secretions such as the nasal secretions, in which high viral RNA copies were measured. Indeed, transmission by contact in humans and pigs was notably reported for other flaviviruses such as dengue, Zika, and Japanese encephalitis virus [11,37,38]. Unfortunately, nasal, oral, fecal, and ocular swab samples were only collected in RNA isolation buffer, which precluded the determination of infectious titers in these samples. Nevertheless, the oral route of transmission via the milk is the most likely based on the higher virus load, the large amount of milk consumed daily by the lambs (200–400 ml), and the extended secretion time. The milk transmission hypothesis is also supported by the behavior of ewes, who normally only allow their own lamb to suckle and charge the other lambs with their heads. If ewes were secreting high infectious viral loads from other body secretions, such as aerosols from the nasal cavity or contaminated pens by the presence of WSLV in feces, we may have seen a more widespread dissemination of the virus among the lambs. Moreover, oronasal virus was detected in lambs only after onset of viremia. On the other hand, the chance for air-borne and contact transmission may have been minimized by higher frequency of air exchange and stable cleaning than in a regular farm environment. WSLV replication in the udder could not be confirmed by immunohistochemical analysis for NS1 staining in the mammary gland of any ewe on 12 dpi. While these data do not formally exclude local virus replication, additional pathways may also contribute to high virus titers in milk including vascular leakage in highly viremic animals, and virus transport within infected immune cells transmigrating from the blood to the milk. Our study design did not permit to test the opposite route of transmission, from lambs to ewes. Indeed, retrograde infection via milking procedures is likely the principal transmission route in the current epidemic of H5N1 highly pathogenic avian influenza clade 2.3.4.4b in dairy cows in United States [39]. Additional studies are necessary to test specific transmission mechanisms.

At necropsy, only few and unspecific macroscopic lesions were observed in animals from both groups, consisting in mild edema in body cavities, which was consistent with previous reports [14,26]. In contrast, virus load in tissues and histological lesions suggest a preferential neurotropic illness in the group inoculated with the clade II SAH177 strain. In contrast, the clade I rSA999 strain induced a more severe viscerotropic disease with a higher virus load in abdominal organs and increased liver and kidney damage. Histological hepatic lesions were

mostly of mild intensity and corresponded to the non-specific pattern of degenerative changes in hepatocytes previously reported [14,26,40], except for the lamb that succumbed to rSA999 infection, which presented severe widespread lesions in not only the liver but also the kidneys. These findings correlated well with the clinical biochemical data of hepatocellular disease. The altered biochemical hepatic parameters are consistent with elevated transaminases reported in the case of a farmer infected with SA999 [21] and with other viral hepatitis, including sheep infected by RVFV [41]. Although most ewes showed a cholangitis due to the concomitant mild fluke infection, the altered biochemical hepatic parameters were not observed in the controls and are therefore attributable to the viral infection [42,43]. Moreover, these parameters were similarly altered in infected lambs, which did not show evidence of parasitosis. Fulminant hepatitis may lead to acute renal failure, which may explain the sudden decline of one lamb in the rSA999 group. Given the high WSLV load detected by RT-PCR in the kidney of this animal, the renal lesions may also have been caused directly by the virus. Only the other viremic lamb from the rSA999 group showed histological lesions in the kidney, which were of mild intensity. It is worth noting that NS1 was detected in the liver of 2 ewes of the rSA999 group, which together with the high levels of liver damage biomarkers suggests that WSLV was still replicating in the liver at 12 dpi. These findings indicate that the liver is a major site of WSLV replication leading to a persisting viremia despite high levels of neutralizing antibodies.

Histopathological examination of the CNS revealed a non-suppurative meningoencephalitis in six animals (including lambs and their mothers) with high viremia with SAH177, but in none of those infected with rSA999. These findings are striking because, to our knowledge, this is the first documented case of WSLV-associated meningoencephalitis in ewes and horizontally infected lambs [14,28]. In previous studies, meningo-encephalitis and brain malformations were only found in aborted and stillborn lambs infected during gestation when ewes were immunized with the live attenuated Ondertespoort WSL vaccine strain, or from ewes that received combined immunization for WSL and Rift Valley fever [24,40]. These data suggest that some WSLV strains may preferentially cause a neurotropic disease. Indeed, the neurotropic phenotype may be due to serial passage in the brains of newborn mice, which was the original method of isolation [12], including for the SAH177 strain used here. Similarly, the attenuated Rift Valley fever Smithburn vaccine strain, generated by serial passage in mouse brain, was more neurotropic and less hepatotropic [44]. Alternatively, enhanced neurotropism may be a feature of clade II strains, which should be further explored. It is likely that limited, key changes in the M and E proteins are sufficient to change the cellular tropism, as reported for yellow fever virus [45]. On this line, genomic diversity in the now-discontinued, heavily brain-passaged FNV yellow fever vaccine strain was associated with the neuroinvasive phenotype [46]. Most human cases of WSL disease are related to mosquito bites or laboratory handling of the virus, resulting in mild to moderate illness, except for an accidental laboratory case, in which severe disease with encephalitis occurred as a result of spraying a virus suspension into the face and eyes [21,47].

Little is known about protective immunity against WSLV, but it is likely that animals and humans who were exposed and survived the infection develop long-term immunity, similar to yellow fever virus. In this study, we have demonstrated that ewes and lambs mount a strong neutralizing antibody response against WSLV within a few days after infection. Additionally, we also found that antibodies were cross-neutralizing rSA999 and SAH177 in *in vitro* assays, indicating that WSLV strains from two clades belong to a unique serotype. Further research is needed to understand protective immune responses against WSLV and whether immunity against related orthoflaviviruses, such as yellow fever virus, lead to cross-protection or, on the contrary, is associated to antibody-dependent enhancement of disease. The development of a

laboratory animal model of WSL disease would help accelerate discovery and therapeutic development.

In conclusion, our work demonstrates that WSLV is found at very high viral titers in the milk of infected ewes, and likely also in the milk of other ruminants, thus indicating that WSLV might constitutes a serious public and animal health hazard in many African regions south of the Sahara. Strong recommendations against feeding raw milk from small ruminants to infants and children should be renewed and endorsed, especially in endemic regions. In a One Health approach, differential laboratory and field diagnostic methods for humans, livestock, and wildlife need to be developed and applied to better identify the emergence of WSLV against clinically similar diseases such as yellow fever or Rift Valley fever. The expansion of WSLV beyond known endemic areas, where competent vectors and naïve hosts are present, also seems to be a clear risk that should be evaluated and monitored thoroughly in insect and serological surveillance programs worldwide.

## Materials and methods

### Ethics statement

The animal experimentation on sheep (*Ovis orientalis aries*), Skudde breed, was conducted within the containment (BSL3Ag), insect-free facilities of the Institute of Virology and Immunology (IVI, Mittelhäusern) in strict compliance with the Swiss animal protection law (Article 18 Animal Welfare Act (SR 455), Article 141 Animal Welfare Ordinance (SR 455.1), and Article 30 Animal Experimentation Ordinance (SR 455.163)), following good animal practice as defined by European regulations, and approved by the Cantonal ethical committee for animal experiments under the license number BE69/2022.

### Cells

Vero cells (ATCC; CCL-81) were cultured in Dulbecco's modified Eagle's medium (DMEM; Gibco) supplemented with 10% fetal bovine serum (FBS; Hyclone), non-essential amino acids (MEM NEAA; Gibco), and 100 units/ml penicillin and 100 μg/ml streptomycin (Gibco). The *Aedes albopictus* C6/36 cells (ATCC; CRL-1660) were cultured in minimum essential medium (MEM; Gibco) supplemented with 100 mM sodium pyruvate (Gibco), non-essential amino acids, and 10% or 2% fetal bovine serum (FBS) (v/v) for cell growth or virus propagation, respectively. The cells were maintained at 37°C (Vero) or 28°C (C6/36) in a 5% $CO_2$ atmosphere.

### TAR cloning and virus rescue

The assembly of full-length rSA999 WSLV cDNA was generated based on a previously established method [48]. Briefly, six overlapping synthetic fragments covering the entire SA999 sequence were designed based on the GenBank sequence (MK163943) and synthetically produced (GenScript). Additional sequences were included at the 5' and 3' ends for cloning. The synthetic fragments were mixed with a pCC1BAC-His3 vector and cotransformed in *S. cerevisiae* VL6-48N for assembly by homologous recombination (S1 Fig). Clones were screened by using a multiplex PCR and restriction analysis with *Mfe*I (New England Biolabs). Plasmid DNA extracted from two selected yeast clones were transformed into *E. coli* EPI300 and subsequent DNA preparations were verified by multiplex PCR, restriction digest, and Sanger-sequenced. One clone showed 100% sequence identity with the expected sequence while the other had a single SNP (g>t) at position 893. The plasmid encoding full-length SA999 WSLV cDNA with 100% identity was first linearized with I-*Sce*I (New England Biolabs) and then

purified by phenol-chloroform extraction and ethanol precipitation. The resulting DNA was used for in vitro transcription using the mMESSAGE mMACHINE T7 Ultra Kit (Thermo Fisher). Approximately 1 ug of transcribed-, and 5'-capped RNA was electroporated into Vero cells. Seventy-two-hours post transfection, the cell supernatant (passage 0 virus) was used for viral propagation in C6/36 cells. The rescued viruses were passaged twice. Viral titers were determined in C6/36 cells using the method described below.

## Virus stocks for in vivo studies

Clade I WSLV SA999 strain (GenBank: MK163943) was identified in South Africa and we generated infectious clones by reverse genetics using TAR cloning (see above). rSA999 stock virus was produced in C6/36 cells and used at passage 2. Clade II WSLV SAH177 99871–2 strain was provided by European virus archive global (EVAg, Marseille, France). SAH177 was originally isolated from a human subject in 1955 in South Africa [27], and passaged 21 times in suckling mice. It was donated by the University of Texas Medical Branch to the EVAg, where it was passaged at least 6 times in C6/36 cells (GenBank: DQ859058.1) [29]. The virus was passaged 3 times in C6/36 cells at the IVI to generate the working stock. Whole genome sequencing of the virus working stocks was performed using a MinION Mk1B (Oxford Nanopore Technologies). Briefly, library preparation and barcoding were performed using the Rapid Barcoding Kit 96 (SQK-RBK114.96), Oxford Nanopore Technologies). Data acquisition and real-time high-accuracy base-calling was performed using MinION software (version 24.2.16). Geneious Prime 2024.0.4 was used to map minimap2 fastq pass reads to the reference sequences and generate consensus sequences of each stock. Details of the rSA999 sequence assembly and the consensus sequence are shown in S1 and S3 Appendix, respectively. Details of the SAH177 sequence assembly and the consensus sequence are shown in S2 and S4 Appendix, respectively.

## Animal experimentation

A total of 27 Skudde breed sheep were included in this experiment with the primary aim being to determine the presence of WSLV in the body fluids of inoculated ewes. Prior to experimentation, the minimum group size was calculated assuming that WSLV RNA would increase in 90% of the inoculated animals with a power of 0.8, having a standard error below 40%, and assuming an error of 0.05. Therefore, at least three animals per group were required. Animals were randomized into groups based on the order of birth. The ewes and their offspring were successively assigned to the mock control, and the two infected groups. Priority was given to selecting ewes with only one lamb, when not, the ewe with two lambs was randomly assigned to one group to be infected. Therefore, the non-infected (mock control) group was composed of three lactating ewes (n = 3) and their corresponding lambs (one per each ewe; n = 3). For each of the WSLV-challenged groups, five lactating ewes (n = 5) and their corresponding lambs were included (n = 5 lambs for the rSA999 group; n = 6 lambs for the SAH177 group). Details about the age, origin, and lambing history of the ewes and inoculation day relative to birth the lambs are detailed in S1 Table. The lambs acted as sentinel animals for detecting an eventual horizontal transmission. Ewes and their newborn lambs were moved from the on-site breeding ground to the BSL3Ag stables to acclimate at least 2 days prior to inoculation. Each group was housed in a separate BSL3Ag isolation stable. WSLV inoculation was blinded with respect to the virus strain. Ewes were intravenously inoculated via the jugular vein with $10^5$ TCID$_{50}$/animal in 1 ml of MEM with either the clade I rSA999 WSLV strain or the clade II SAH177 WSLV strain. The mock group received only 1 ml of MEM with cell culture supernatant (Fig 1A). During the experiment, clinical evaluation, including measuring body

temperature and assessing awareness, appetite, manure excretion, breathing, gait, and neurological signs, was performed for all the animals by the same veterinarian. Additionally, the body weight of the lambs was recorded daily. Milk from ewes and blood samples from all animals in the infected groups were collected daily, while samples from the mock group were collected every other day starting one day prior to inoculation. Oronasal swabs (FLOQSwabs; COPAN) were collected from lambs, and swabs from the nasal cavity, eye conjunctiva, oral cavity, and rectum were collected from ewes until day 11 of the study. Nasal and conjunctival swabs were taken alternatively from the left and right sides. Immediately after collection, swabs were immersed in 750 µl of RA1 lysis buffer (Macherey-Nagel; Germany) and stored at -70˚C until further analysis. On day 12 of the experiment, sheep were euthanized by exsanguination under deep narcosis (xylazine/ketamine). Blood and organs were collected for further examination.

### RT-qPCR for viral RNA

Immediately after euthanasia, 3–5 mm$^3$ organ samples, including olfactory bulb, cortex, thalamus, cerebellum, tonsil, mandibular and inguinal lymph nodes, spleen, kidney, liver, mammary gland (ewes only), and thymus (lambs only) were collected in 1.5-ml tubes (Sarstedt) containing 750 µl of lysis buffer RA1 (Macherey-Nagel), then homogenized using a BulletBlender (Next Advanced Inc, USA), weighed, and frozen at -70˚C. Then, 200 µl of each sample (serum, milk, RA1 buffer from swabs, or RA1 buffer from centrifuged lysed organs) were used for viral RNA extraction performed with the NucleoMag VET kit (Macherey-Nagel) and the extraction Kingfisher Flex robot (ThermoFisher Scientific). Viral RNA was amplified by RT-qPCR using the AgPath-ID One-Step RT-PCR kit (Applied Biosystems) and primers designed to target the highly conserved 3'-UTR region of WSLV. A 162 bp amplicon was generated with forward 5'-TTACCGCGCACGGTGGGAAA-3' and reverse 5'-CCCGCCCTGCATTCAAG-CAA-3' primers in combination with a fluorescent probe 5'-FAM-CCCTCCCGAGCACACA-TAGCGGACC-BQH1-3'. The viral genome equivalents were determined using a standard curve created by ten-fold serial dilutions of a *Not*I linearized pCR4-TOPO TA plasmid containing the cloned 162bp target WSLV amplicon with a detection limit set at 10 copies/µl per reaction (2 µl). RT-qPCR and subsequent analyses were performed using the QuantStudio 5 Real-Time PCR system (ThermoFisher Scientific).

### Virus titration

Viral titers were determined on C6/36 or Vero cells using the same cell type as the virus stocks used in the assay. For milk samples, the viral loads were determined in C6/36 cells. For the samples, a one-in-ten dilution was prepared in cell-type-specific medium supplemented with 2% FBS (v/v) as previously mentioned, for milk samples, with 1% antibiotics (penicillin-streptomycin; Gibco), further ten-fold dilutions were performed, and incubated on cells for 72 h at either 37˚C for Vero cells or 28˚C for C6/36 cells. In a similar manner, viral plaque-forming units were determined on Vero cells. For this, ten-fold serially diluted virus stocks were added to the cells, incubated for 2 h at 37˚C. After washing with warm D-PBS, a semifluid Vero cell medium supplemented with 1% methylcellulose (Merck) was added. After washing the supernatants or the viscous overlay, cells were fixed with a 4% formalin solution. Thereafter, infected cells were identified using the immunoperoxidase monolayer assay (IPMA) with the anti-Flavivirus E monoclonal antibody 4G2 (ATCC; HB-112). Then, viral titers from supernatants and milk samples were calculated using the Reed and Muench method [49] and expressed as 50% tissue culture infective dose per ml (TCID$_{50}$/ml). For plaque assays, the viral plaques were

counted at the highest dilution. The viral titer was then calculated by applying the dilution factor and expressed as plaque-forming units per ml (PFU/ml).

## Virus serum neutralization test

Serum samples were heat-inactivated at 56˚C for 30 minutes. Then, sera from all animals at every day, and milk from ewes on indicated days (3, 4, 5 and 10 dpi). The samples were two-fold serially diluted in Vero cell medium supplemented with 2% FBS and, for milk samples, with 1% antibiotics (penicillin-streptomycin; Gibco). All dilutions started at 1:10 (sample: medium) in a total volume of 100 µl. Then, 200 FFU/well of WSLV rSA999 or SAH177 strain were added to the diluted samples, making a total volume of 200 µl. These viral stocks were produced and titrated using Vero cells. After 1 h at 37˚C, 100 µl of the mixture containing 100 WSLV PFU were added to confluent Vero cells plated in 96-well plates. The cells were then incubated for 72h at 37˚C with 5% $CO_2$. Afterward, the supernatants were removed, the cells were washed once with warm D-PBS, and then fixed for 10 min with a 4% formalin solution at room temperature. Infected cells were identified by IPMA staining of the E protein, as previously described. Next, plates were imaged using the Immunospot machine (Cellular Technology LTD), and the area of E-positive cells for each well was determined with Image J software. The use of naïve serum as a negative control enabled us to assess the extent of E protein staining per well infected with 100 WSLV PFU, establishing the baseline level for the test at 100%. The presence of neutralizing antibodies in the immune serum reduces or completely prevents cell infection, leading to a decrease in E protein staining on the well. Taking this into consideration, we identified the sample dilution that resulted in at least a 50% reduction in E protein staining in a specific well. Then, the 50% neutralization dose ($ND_{50}$) was calculated using the Spearman-Kaerber formula [50].

## Serum biochemistry analytes

Aliquots of serum samples were thawed and treated with Nonidet P-40 (NP-40 at 0.5% v/v; ThermoFisher Scientific) and incubated for 1 h at room temperature for virus inactivation [51]. Samples were then stored at -80˚C, or on dry ice during shipping, until analyzed at the Interdisciplinary Laboratory of Clinical Analysis (Interlab-UMU) of the University of Murcia (Spain) on an automated biochemical analyzer Olympus A400 (Beckman Coulter). Triglycerides, cholesterol, Alkaline Phosphatase (ALP), Aspartate Transferase (AST), blood urea nitrogen (BUN), creatinine, bile acids, bilirubin, glucose, and lactate were measured using commercial kits from Beckman Coulter; total protein and albumin concentrations were measured using Olympus Life and Material Science; adenosine deaminase (ADA) was determined by spectrophotometry using a commercial kit (Diazyme Laboratories) adapted for use in an automated analyzer [52].

## Histopathology and immunohistochemistry

Half of the brain (including one complete cerebral and cerebellar hemisphere), cervical spinal cord, tonsil, lymph nodes (mandibular, tracheobronchial, inguinal and mesenteric), spleen, kidney, liver, ileum, heart from all animals, as well as mammary gland and thymus from the ewes and lambs, respectively, were fixed in 4% buffered formalin for two weeks. Subsequently, cross-sections of all collected tissues were performed, including representative areas of the CNS (olfactory bulb, frontal lobe, basal nuclei, hippocampus, cerebral cortex, thalamus, midbrain, pons, cerebellum and cervical spinal cord), and were embedded in paraffin, cut at 4 µm of thickness and stained with hematoxylin and eosin (H&E) for histological examination.

Lesions were semi-quantitatively scored by a board-certified pathologist (LGR) from 0 to 3 as follows: 0: none; 1: mild; 2: moderate; and 3: severe.

For IHC, 2 μm formalin-fixed, paraffin-embedded tissue sections were mounted on positively charged slides, dried for 30 min at 60˚C and subsequently dewaxed, pretreated and stained on an Immunostainer Leica Bond RX (Leica Biosystems). After dewaxing (Bond Dewax solution; Leica Biosystems), slides were incubated in a Citrate buffer (Bond Epitope Retrieval 1; pH 6) for 30 min at 100˚C, using a protein block (FBS 1%) to reduce non-specific binding of the primary antibody. The slides were then incubated with a primary mouse monoclonal antibody against WSLV non-structural protein 1 (NS1) (HM1022, Medix Biochemica) at 1:100 for 60 min at room temperature. All further steps were performed using reagents of the Bond Polymer Refine Detection Kit (Leica Biosystems) as follows: Endogenous peroxidase was blocked for 5 min, then a rabbit-anti-mouse secondary antibody was applied (15 min), followed by a peroxidase-labelled polymer (15 min). Finally, slides were developed in 3,3'-diaminobenzidine/$H_2O_2$ (10 min), counterstained with haematoxylin, and mounted. The IHC was semi-quantitatively assessed as follows: 0: negative; 1: positivity in few hepatocytes (<5%); 2: positivity in 5–20% hepatocytes; 3: positivity in >20% hepatocytes. The IHC for WSLV was performed in the liver samples as well as in all collected tissues from selected animals (ewe represented by an inverted triangle from the rSA999 group, lamb represented with a rhombus and a dot from the SAH177 group, and ewe represented by a circle from the control group).

## Statistics

GraphPad Prism 10 Software (GraphPad Software, Inc., La Jolla, San Diego, CA, USA) was used for data analysis and generation of figures. For each group of ewes or lambs, the area under the curve (AUC) was calculated for every measured parameter throughout the entire duration of the experimental period. Differences between groups were subsequently determined using a one-way ANOVA with Tukey *post hoc* test for multiple comparisons. For the daily body weight gain, individual daily differences were determined using the Mann–Whitney U-test with Bonferroni *post hoc* test. Correlation analysis between WSLV RNA levels in milk and viral titers was calculated by Spearman's Rho analysis; the correlation was considered relevant with a $R^2$ >0.5. A p<0.05 was considered statistically significant, as indicated in the figures by asterisks: *p < 0.05, **p ≤ 0.01, ***p ≤ 0.001, and ****p ≤ 0.0001.

## Supporting information

**S1 Fig. TAR cloning and rescue of WSLV SA999 strain.** (**A**) Schematic representation of the 6 synthetic overlapping DNA fragments (FG1-FG6) spanning the whole genome of WSLV SA999 virus sequence and including anchors for cloning into pCC1BAC-His3 vector. (**B**) Map of the cloned SA999 genome. (**C**) Multistep process to generate infectious WSLV rSA999 virus particles (see methods for details). The figure was created using BioRender. (PNG)

**S2 Fig. Serum concentration of additional biomarkers in WSLV-infected sheep.** (**A**) Alkaline phosphatase (ALP), (**B**) albumin, (**C**) glucose, (**D**) lactate, (**E**) triglycerides, and (**F**) total proteins. The maximum and minimum values from the mock group of ewes and lambs were indicated for reference in gray shading background. (**G**) Group differences were analyzed by one-way ANOVA of the AUC with Tukey *post hoc* test. An asterisk on a specific group indicates differences compared to the age-matched mock; underlined asterisks indicate differences between infected groups. Statistical significance was considered for p<0.05

(***p<0.001; **** p<0.0001).
(TIF)

**S1 Table. Experimental animals metadata.**
(XLSX)

**S1 Appendix. Sequencing report WSLV rSA999.**
(PDF)

**S2 Appendix. Sequencing report WSLV SAH177.**
(PDF)

**S3 Appendix. GenBank consensus sequence file for WSLV rSA999 working stock.**
(GB)

**S4 Appendix. GenBank consensus sequence file for WSLV SAH177 working stock.**
(GB)

**S5 Appendix. Data Tables.**
(XLSX)

## Acknowledgments

We thank Katarzyna Sliz, Daniel Brechbühl, Roman Troxler, Jan Salchli, and André Rohn for dedicated animal care; we are grateful to the biosafety team of the IVI for support and advice. We thank Marianne Bärtschi, Ursula Reinhart, Doris Röder-Hajek and Brigitte Tenisch from the Institute of Animal Pathology of the University of Bern for their excellent service in histology and Therese Waldburger and Cristina Graham from the Translational Research Unit of the University of Bern for their excellent service in IHC. We thank Brooke Mitchell from the University of Texas Medical Branch for providing details on the passage history of the SAH177 strain.

## Author Contributions

**Conceptualization:** Matthias Liniger, Artur Summerfield, Nicolas Ruggli, Charaf Benarafa, Obdulio García-Nicolás.

**Data curation:** Marta Zimoch, Llorenç Grau-Roma, José Joaqín Cerón, Charaf Benarafa, Obdulio García-Nicolás.

**Formal analysis:** Marta Zimoch, Llorenç Grau-Roma, Damián Escribano, Paraskevi Pramateftaki, Sergi Torres-Puig, Charaf Benarafa, Obdulio García-Nicolás.

**Funding acquisition:** Artur Summerfield, Charaf Benarafa, Obdulio García-Nicolás.

**Investigation:** Marta Zimoch, Llorenç Grau-Roma, Noelle Donzé, Aurélie Godel, Damián Escribano, Bettina Salome Trüeb, Paraskevi Pramateftaki, Sergi Torres-Puig, Nicolas Ruggli, Charaf Benarafa, Obdulio García-Nicolás.

**Methodology:** Marta Zimoch, Llorenç Grau-Roma, Matthias Liniger, Damián Escribano, Bettina Salome Trüeb, Paraskevi Pramateftaki, Sergi Torres-Puig, José Joaqín Cerón, Volker Thiel, Jörg Jores, Nicolas Ruggli, Charaf Benarafa, Obdulio García-Nicolás.

**Project administration:** Charaf Benarafa, Obdulio García-Nicolás.

**Resources:** Jörg Jores, Artur Summerfield, Nicolas Ruggli, Charaf Benarafa, Obdulio García-Nicolás.

**Supervision:** Charaf Benarafa, Obdulio García-Nicolás.

**Validation:** Charaf Benarafa, Obdulio García-Nicolás.

**Visualization:** Marta Zimoch, Llorenç Grau-Roma, Charaf Benarafa, Obdulio García-Nicolás.

**Writing – original draft:** Charaf Benarafa, Obdulio García-Nicolás.

**Writing – review & editing:** Marta Zimoch, Llorenç Grau-Roma, Matthias Liniger, Jörg Jores, Charaf Benarafa, Obdulio García-Nicolás.

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
