## [Decision Letter · Decision Letter 0]

7 Oct 2024

Dear Prof. Dr. Benarafa,

Thank you very much for submitting your manuscript "Mosquito-free milk-associated transmission of zoonotic Wesselsbron virus in sheep" for consideration at PLOS Pathogens. As with all papers reviewed by the journal, your manuscript was reviewed by members of the editorial board and by several independent reviewers. The reviewers appreciated the attention to an important topic. Based on the reviews, we are likely to accept this manuscript for publication, providing that you modify the manuscript according to the review recommendations.

Sincerely,

Amy L. Hartman, PhD

Academic Editor

PLOS Pathogens

Ashley St. John

Section Editor

PLOS Pathogens

Michael Malim

Editor-in-Chief

PLOS Pathogens

orcid.org/0000-0002-7699-2064

Reviewer Comments (if any, and for reference):

Reviewer's Responses to Questions

**Part I - Summary**

Reviewer #1: The manuscript entitled "Mosquito-free milk-associated transmission of zoonotic Wesselsbron virus in sheep”, describes an interesting, primary descriptive, study in which an extensive data package is presented in a sound and conclusive way that sheds new light on mosquito-independent (vertical) transmission of WSLV, most likely mediated by milk. The unique experiment paves the way for similar studies with other arboviruses that may use this route as well and opens opportunities for more mechanistical/fundamental study approaches. This being said, the manuscript could be improved by addressing the following points.

Reviewer #2: The authors conducted a very interesting and novel study by inoculating ewes with WSLV from two different clades. Instead of pregnant ewes, this study utilized ewes who had nursing young lambs. The study provides a wealth of information about viral RNA copies in different tissues and secretions, clinical biochemistry, macroscopic pathology, histopathology and neutralizing antibodies. All of this is very valuable data for future studies. It is indeed very surprising that WSLV can be transmitted to lambs via milk. It has always been assumed that the virus crosses the placenta or that lambs become infected by nasal secretions from their dam or that transmission is via mosquito bites.

This paper also provides more evidence that Clade II viruses may be neurotropic and Clade I viruses hepatotropic. This has important implications for the diagnosis of WSL disease. Traditionally, only liver samples are tested for WSLV when a case is suspected.

The manuscript is well written and there are only minor grammatical errors.

**Part II – Major Issues: Key Experiments Required for Acceptance**

Reviewer #1: (No Response)

Reviewer #2: No major modifications required.

**Part III – Minor Issues: Editorial and Data Presentation Modifications**

Reviewer #1: 1) Title; it might be better to state mosquito-independent instead of mosquito free.

2) The short title is longer than the normal title

3) The ‘flow’ in the abstract is not optimal and could benefit from some editing.

4) Intro: Line 1: epidemics of what?

5) The age, lambing history and origin of the individual ewes is not reported, nor the randomization strategy. Please add this information. Furthermore, from the results section it is suggested that the 3 groups were housed in different stables though this is not explicitly stated.

6) The section “Virus stocks for in vivo studies” does not read well. Furthermore, it does not becomes clear whether the rescued viruses used to inoculate were confirmed by sequencing.

7) In addition to point 6, have the authors sequenced samples (including milk samples) coming out of the trial to assess for the presence of adaptive mutations?

8) The limit of detection for the PCR is set to 10 genome copies however it can be clearly seen that in the lower ranges, samples tested either 10^3 or negative. Consequently it is very unlikely that the lower limit of detection is 10. Please explain and/or adapt.

9) Fig 3: the scale is log 10 though linear values are presented

10) Fig 2 F: there is one sample highly positive at day 5 but related samples at day 3 and 6 seem to be negative. This seems biologically highly unlikely. Could you please explain.

11) Have the authors assessed potential replication site in the udder, or how would the virus end up in the milk?

12) The horizontal/vertical transmission route as presented might also impact transmission dynamics from an epidemiology view. With one mosquito multiple animals may become infection (one ewe and its lambs). Have the authors fed this data into transmission models?

13) Could transmission also be the other way around. From lambs to ewes. The dataset possibly does not allow any conclusions here but this could be discusses in the discussion section.

Reviewer #2: INTRODUCTION:

Remove the word ‘However’ from this sentence or replace with ‘For instance’: However, the outbreaks of Zika virus (ZIKV) in South America and West Nile virus (WNV) in North America……

Regarding this sentence: Natural outbreaks of WSL disease are clinically similar to Rift Valley fever with abortions and death of newborn lambs, which requires proper diagnostic confirmation that is often absent in resource poor regions of Africa.

It is not that diagnostic services are absent. There is in fact fairly good diagnostic services in many African countries. It is more that lambs die from so many causes that farmers often don’t attempt to have a diagnosis made. Sheep are also often farmed extensively in arid areas, and abortions or lamb deaths are not detected soon enough to make a diagnosis possible. Fetuses with malformation are sometimes presented for necropsies but the diagnosis cannot be made at this stage since viral RNA is no longer detectible in the tissues. This study also reiterates that the macroscopic lesions and histological lesions are unremarkable in most cases.

Rather something like….. which requires appropriate sampling and multiple diagnostic techniques to confirm the diagnosis. Or …..can be challenging to diagnose since lesions are unremarkable or non-specific in most cases.

RESULTS:

PAGE 4, PARAGRAPH 2: This sentence: A more recent study using the clade I strain SAH177 [27], which was isolated from a human and passaged in mouse brains and Vero cells, reported early pathogenesis of WSLV infection in pregnant ewes [28].

I believe SAH177 is a clade II strain. Reference 27 also seems to in the wrong place. Passaged in mouse and Vero cells in 1957 [27], reported…..

There is not much is mentioned about the pathogenicity of clade II strains in this paragraph. I suggest adding a line of 2 about this.

It seems that not much is known about the pathogenicity of SA999 (clade II) in sheep as well. However, the VanTonder strain, also in clade II, was inoculated into 30 pregnant ewes [Weiss et al] of which 4 gave birth to weak lambs that died or were killed in extremis at 2 to 3 dpi and 8 aborted between 4 and 7 dpi. Six of the ewes died at which point the experiment was terminated due to biosafety concerns.

PAGE 6, UNDER THE HEADING: High WSLV titers in milk of infected ewes correlates with transmission to lambs

The overlapping lines in Figure 2C is difficult to interpret. Many of the symbols overlap making it impossible to tease data for different animals apart. Split this graph into 2 graphs like the 2 graphs for the serum samples.

PAGE 7, VIRAL LOAD IN ORGANS:

The description of how many organs was positive for the ewes inoculated with SAH177 is not clear. Rewrite to make the meaning clear.

This sentence: The ewe with the shortest viremia was RT-qPCR negative in the rest of tested organs, whereas the four other ewes had detectable virus in the tonsils, and in one or two additional samples including spleen, lymph nodes, kidneys, and mammary gland.

From the graph in Figure 4 it seems that 3 ewes (and not 4 as stated) had virus in the tonsils and 3 had virus in the spleen (not one or 2 as stated). It is also difficult to interpret the data in the graphs.

PAGE 10, FIRST PARAGRAPH: This sentence: Moreover, results suggest that the two strains may differ in their tissue tropism preferences: neurologic (SAH177) and hepatic (SA999).

This is a very important finding. Brain samples are rarely submitted for HE and IHC for WSL diagnosis whereas liver samples are routinely submitted. It seems from this study that at least one ewe and one lamb only had lesions in the CNS and not in the liver. However, it also seems that a variety of organs (including brain samples) were positive using RT-qPCR. This implies that brain samples should be routinely submitted to detect infections with clade II viruses and that RT-qPCR should also be routinely conducted on these samples since the lesions are often non-specific and IHC negative.

PAGE 11, FIRST PARAGRAPH: This sentence: Similarly, TBEV is secreted……

Write TBEV out since there does not seem to be any mention of this virus elsewhere in the text.

JEV as well in the next paragraph.

Rift Valley fever and yellow fever was written out elsewhere. Either write all names of viruses out or use abbreviations consistently. However, it is easier to follow text if names that are not often used are spelt out.

Page 1, last paragraph: ‘….inoculated with the clade I SAH177 strain. In contrast, the clade II rSA999 strain induced a more severe…..’

That should be clade II SAH177 strain and clade I rSA999 strain.

PAGE 11, LAST SENTENCE:

The paper by Le Roux (1959, The histopathology of Wesselsbron disease in sheep) should be read and cited here. The histopathology of the liver in natural Wesselsbron cases can indeed be difficult to interpret (non-specific). There is no typical pattern for degenerative changes in hepatocytes which may be centrilobular, periportal or diffuse. IHC positive cases sometimes only have rare hypereosinophilic, shrunken hepatocytes with rare IHC positive hepatocytes and Kupfer cells. Other cases have diffuse necrosis that looks very similar to RVFV infection in lambs. The only consistent finding seems to be proliferation of Kupffer cells. Unfortunately, natural cases are rarely found, and the histopathology still requires more study.

PAGE 12, SECOND PARAGRAPH: This sentence: These findings are striking because there is only little evidence of encephalitis experimentally infected animals, which is has been described mostly in newborn and aborted lambs [14, 28].

Remove this sentence or modify.

Coetzer et al 1978 [your reference 14] distinctly mentions that encephalitis was not found in any of the brain specimens. Oymans et al. 2020 only mentions strong staining of WSLV antigen throughout the fetal brain that included differentiated neurons, neuroglial cells and microglial cells with the formation of microglial nodules. I think the authors of the new research can claim that encephalitis with neuronal cell death accompanied perivascular cuffing is a finding that has not been described before (unless the authors can point a paper out that I missed). This has important implications for diagnostics. A recommendation should be added that when Wesselsbron is suspected, brain samples should be collected in addition to liver samples, from multiple location and histology and RT-qCR done. Can the authors recommend which areas the brain they consider critical or most useful for sampling?

Also, describe the origin of the SAH177 strain better. A field worker became ill with what was assumed to be encephalitis. No symptoms were described that suggests hepatic disease. Therefore, it is assumed that this virus, and by extension the Clade II viruses, are neurotropic. I also suggest that the authors mention the Onderstepoort Wesselsbron vaccine strain that causes malformations of the CNS in sheep fetuses. Also note that RVFV becomes neurotrophic when passaged in the brains of new-born mice. However, I am not sure if the original stain used to produce Smithburn vaccine was determined to be neurotropic rather than hepatotropic. This might be interesting to add.

FIGURE 2C: Too many overlapping lines and symbols. Split into 2 separate graphs for rSA999 and SAH177.

FIGURE 4: Put the data in tables instead of graphs. It is very difficult figure out which organ was positive in each animal.

FIGURE 6: Lengthen the arrow heads so that the tip points exactly to the lesion. At the moment they are triangles and one has to guess which tip of the triangle points to the lesion.

PLOS authors have the option to publish the peer review history of their article (what does this mean?). If published, this will include your full peer review and any attached files.

Reviewer #1: No

Reviewer #2: **Yes: **Prof Lieza Odendaal

Figure Files:

Data Requirements:

Reproducibility:

References:

---

## [Editor Report · Decision Letter 1]

13 Nov 2024

Dear Prof. Dr. Benarafa,

We are pleased to inform you that your manuscript 'Mosquito-independent milk-associated transmission of zoonotic Wesselsbron virus in sheep' has been provisionally accepted for publication in PLOS Pathogens.

Best regards,

Amy L. Hartman, PhD

Academic Editor

PLOS Pathogens

Ashley St. John

Section Editor

PLOS Pathogens

Michael Malim

Editor-in-Chief

PLOS Pathogens

orcid.org/0000-0002-7699-2064
---

## [Editor Report · Acceptance letter]

30 Nov 2024

Dear Prof. Dr. Benarafa,

We are delighted to inform you that your manuscript, "Mosquito-independent milk-associated transmission of zoonotic Wesselsbron virus in sheep," has been formally accepted for publication in PLOS Pathogens.

Best regards,

Michael Malim

Editor-in-Chief

PLOS Pathogens

orcid.org/0000-0002-7699-2064